# Large language models for psychosocial risk assessment: A multi-method evaluation across suicide, intimate partner violence, and substance misuse

Laura M. Vowels [1,2]*, Pranika Vohra[3], Danyang Li[4], Pegah Zeinoddin[5], Alex Elswick[6], Tiffany Marcantonio[7], Nathan D. Wood[8], Matthew J. Vowels[9,10]

1 Institute of Psychology, University of Lausanne, Lausanne, Switzerland, 2 School of Psychology, University of Roehampton, London, United Kingdom, 3 Department of Psychology, North Dakota State University, Fargo, North Dakota, United States of America, 4 School of Psychology, University of Bristol, Bristol, United Kingdom, 5 Institute of Psychology, University of Lausanne, Lausanne, Switzerland, 6 School of Human Environmental Sciences, University of Kentucky, Lexington, Kentucky, United States of America, 7 Department of Health Science, University of Alabama, Tuscaloosa, Alabama, United States of America, 8 Department of Family Sciences, University of Kentucky, Lexington, Kentucky, United States of America, 9 The Sense Innovation and Research Center, Lausanne and Sion, Switzerland, 10 Kivira Health, New York, New York, United States of America

* laura.vowels@roehampton.ac.uk

## Abstract

Psychosocial risk assessment is a cornerstone of mental health care, yet remains resource-intensive and inconsistently delivered across domains such as suicide, intimate partner violence (IPV), and substance misuse. Recent advances in large language models (LLMs) raise the possibility of scalable, conversational agents capable of detecting and evaluating psychosocial risk. Across three interlinked studies, we evaluated the performance of LLMs in this context. Study 1 benchmarked GPT-4 and Claude 3 Sonnet against vignettes constructed from participants' lived-experience, finding high accuracy in detecting risk domains and substantial agreement with participant-rated severity, though suicidality proved more challenging than IPV or substance misuse. Study 2 examined participants' perceptions of LLM-generated responses, revealing that most judged them accurate, empathic, and clinically useful, with no differences across models or domains. Study 3 implemented a supervised, three-agent GPT-4o-based chatbot system integrating one chatbot as a therapeutic agent, a supervisor for risk detection, and a JSON-based assessor for structured evaluation. The therapeutic agent chatbot was successfully completed full risk assessments most of the time while maintaining therapeutic quality. Together, these studies suggest that LLMs can contribute to psychosocial risk detection and structured assessment under controlled conditions, while underscoring the need for careful supervision, rigorous validation, and clearly defined boundaries before consideration of real-world clinical deployment.

**Data availability statement:** The data and materials are available on the Open Science Framework project page: https://osf.io/bhdz5.

**Funding:** The research was funded by the Institute of Psychology at University of Lausanne, Switzerland (to LMV). The funders had no role in study design, data collection and analysis, decision to publish, or preparation of the manuscript.

**Competing interests:** The authors have declared that no competing interests exist.

## Author summary

Assessing psychosocial risk, including suicidality, intimate partner violence, and substance misuse, is a critical but time- and resource-intensive aspect of mental health care. In this research, we explored whether artificial intelligence, specifically large language models (LLMs), could help support this process. Across three studies, we evaluated how well these models can identify and assess risk in realistic scenarios and how people perceive their responses. First, we tested whether LLMs could detect different types of psychosocial risk from real-life–based case descriptions. The models were generally accurate and aligned well with how individuals rated their own experiences, although identifying suicide risk remained more difficult. Second, we asked participants to evaluate the quality of responses generated by the models. Most found them to be accurate, empathic, and helpful. Finally, we developed a multi-agent AI system combining a therapeutic chatbot with additional components for supervision and structured risk assessment. This system was able to complete comprehensive risk assessments while maintaining supportive interactions. Overall, our findings suggest that AI could play a supportive role in psychosocial risk assessment, provided it is carefully designed, validated, and used alongside appropriate human oversight.

Mental health conditions affect more than a billion people worldwide each year, with depression consistently identified as a leading cause of disability by the World Health Organization [1]. In the United States alone, nearly a quarter of adults (22.8%) experienced a mental health condition in 2021, yet more than half did not receive formal treatment [2]. This treatment gap has spurred interest in scalable, cost-effective interventions that can be delivered outside of traditional clinical settings. Digital health interventions, including websites, smartphone applications, and chatbots, have emerged as potential tools to address these needs [1].

Within mental health services, a critical component of both prevention and intervention is risk assessment: the systematic evaluation of behaviors or circumstances that may lead to harm to self or others. Clinical risk assessment informs treatment planning, resource allocation, and safety management. These assessments integrate information from interviews, psychological assessments, medical history, and social context to identify risk and protective factors across domains such as suicidality, intimate partner violence (IPV), and substance misuse. In practice, clinicians rely on a mixture of professional judgment, structured protocols, and actuarial tools, yet even these methods face well-documented challenges around reliability, predictive power, and ecological validity [3,4]. An important factor underlying the aforementioned challenges is that risk is often disclosed by clients indirectly, with cues embedded in vague or contextual language rather than explicit statements [5]. This presents difficulties not only for clinicians but also for digital systems that attempt to approximate human judgment.

Theoretically, risk assessment occupies a unique position at the intersection of clinical decision-making and interpersonal communication. Risk assessment is both a technical process—structured by checklists, guidelines, and protocols—and a relational one, shaped by empathy, trust, and the dynamics of disclosure. Any effort to digitize or automate aspects of risk assessment must therefore grapple with both dimensions. This paper situates itself at that intersection, examining the potential and limitations of large language models (LLMs) to contribute meaningfully to psychosocial risk assessment in ways that are systematic, replicable, and clinically relevant.

## Digital health interventions and their limitations

Digital health interventions have proliferated in recent years, with estimates suggesting that more than 20,000 mental health apps are available globally [6]. Despite this, sustained engagement is poor: fewer than 4% of individuals who download a mental health app continue to use it after 15 days [7]. This drop-off may occur due to lack of personalization, limited therapeutic alliance, and insufficient evidence-based design [1]. From a theoretical standpoint, engagement with digital interventions is often explained through models of therapeutic alliance and social presence, which suggest that when users feel understood, supported, and connected, even by a digital system, they are more likely to remain engaged [8].

Chatbots, in particular, have been proposed as a partial solution to these engagement problems as they can emulate human language and dialogue, which may increase perceptions that one is being heard and listened to in a digital health intervention. By simulating dialogue, users are more likely to perceive a relational connection because chatbots mirror some aspects of human interaction. Systematic reviews and meta-analyses consistently show that chatbots can support users in managing stress, anxiety, and depression, and may even foster therapeutic alliance in ways that encourage treatment adherence [8–13]. However, most chatbots deployed in mental health contexts to date have relied on rule-based architectures with predefined dialogue options, limiting their ability to respond flexibly or detect nuanced cues of risk.

## The rise of LLMs in conversational AI

The invention of Eliza in 1966 marked the beginning of conversational AI, but for decades chatbot development was constrained by decision tree and rule-based approaches, which relied on predefined scripts and conditional 'if–then' rules to generate responses rather than genuine language understanding [14]. The recent rise of large language models has fundamentally transformed this landscape. Unlike earlier systems, LLMs are trained on a massive corpus of text, enabling them to generate contextually relevant, human-like responses. Between 2019 and 2022, the proportion of chatbots powered by LLMs increased from 7.5% to 56%. The public release of OpenAI's ChatGPT in November 2022 represented a tipping point, reaching 100 million users within two months [15]. This surge in use has spurred widespread discussion of LLMs' potential applications in healthcare, with particular attention to their role in providing mental health support [16].

Early evidence suggests that LLM-based chatbots can convincingly simulate therapeutic dialogue as users often perceive their responses as more empathic and helpful than those of clinicians [17–19]. Experimental applications include ChatGPT-assisted therapy for psychiatric inpatients, which improved patients' responses on the World Health Organization's Quality of Life index [20], and GPT-4-based journaling tools that alleviated depression, anxiety, and loneliness [21,22]. A single-session intervention with the GPT-4o-based chatbot Amanda improved well-being and relationship satisfaction while reducing distress [23–25]. These findings intersect with longstanding theories of human–computer interaction, including the "computers as social actors" framework, which demonstrates that people respond to machines using the same social rules they apply to humans [26].

Yet evidence also highlights critical limitations of chatbots. For example, Vowels et al. [19] found that Amanda accurately identified risk in only a minority of cases that a human coder did, raising concerns about adequacy in high-stakes contexts. For example, participant talking about severe anxiety was identified by a human coder as something to assess whereas the chatbot did not prompt the user further. Elyoseph and Levkovich, [27] reported that ChatGPT tended to underestimate suicide severity relative to healthcare professionals based on their ratings of risk vignettes. While

Ghanadian et al. [28] showed high accuracy in detecting explicit suicide risk, the challenge of handling subtle or ambiguous disclosures remains. Together, these findings underscore that while LLMs may enhance engagement and relational connection, their reliability in risk assessment—a task where false negatives can have severe consequences—remains uncertain.

### Risk assessment across domains: suicide, IPV, and substance misuse

Clinical risk assessment is fundamental across several domains. In suicide and self-harm, widely used tools include the Columbia–Suicide Severity Rating Scale, the Suicide Ideation Questionnaire, and the Beck Scale for Suicide Ideation [29]. Yet such instruments may omit contextual factors such as social isolation, economic instability, or recent life changes [5,30]. IPV, affecting one in four women and one in nine men in the United States [31], includes psychological, physical, and sexual abuse, as well as coercive control. Although validated tools exist to identify the presence of IPV [32,33], their predictive consistency is limited [3,34]. Substance misuse, encompassing harmful use of alcohol, illicit drugs, and prescription medications, poses further challenges, contributing to an estimated $420 billion in annual societal costs in the United States and increasing risk for suicide, self-harm, and IPV [35].

Digital applications aimed at risk assessment in these domains often fall short. Suicide prevention apps rarely conduct active assessments or include accurate crisis resources [36,37]. IPV-focused apps seldom integrate emergency assistance, avoidance strategies, or legal information [38]. Substance misuse apps frequently limit their scope to alcohol without addressing poly-substance use [39,40]. Even chatbot-enabled tools remain constrained, often relying on rule-based architectures or failing to detect indirect expressions of risk (e.g., 'I feel very sad and hopeless' [37,41]). These shortcomings reflect broader critiques of digital health interventions, which emphasize gaps between technical innovation and clinically meaningful impact [42].

By focusing on suicidality/self-harm, IPV, and substance misuse, the present research targets three domains that are highly prevalent, clinically consequential, and theoretically distinct. Together they represent internalizing, interpersonal, and behavioral risks, illustrating different patterns of disclosure and requiring varied forms of assessment. Importantly, they also frequently co-occur, amplifying overall risk. Evaluating LLM performance across these domains provides a broader and more clinically relevant test than focusing on each risk in isolation.

### Existing evidence and remaining gaps

Despite growing enthusiasm, research on LLMs in psychosocial risk assessment remains limited and fragmented. Much of the existing work has focused narrowly on ChatGPT, with most evaluations restricted to a single domain such as suicide [27,28]. Other studies have examined adjacent sensitive areas, such as sexual violence, and shown that with carefully structured prompts, models like GPT-4, Claude, Llama, and Mistral can align closely with experts' judgments [43]. These findings highlight promise but are constrained by methodological limitations. Many rely on short, artificial prompts rather than extended, ecologically valid cases. Few studies have included multiple forms of psychosocial risk, and even fewer have compared different models under identical conditions. Importantly, there is still no established framework to evaluate whether LLMs can not only detect risk but also produce assessments that capture severity, contextual factors, and appropriate recommendations in ways consistent with clinical reasoning. Even when overall detection accuracy appears high, failure cases may be clinically significant, as errors often occur in precisely the kinds of ambiguous or high-stakes disclosures that matter most. Understanding the nature and context of these misclassifications is therefore as important as measuring accuracy itself, and vignette-based approaches such as those used in the present study provide a means to examine and characterize these patterns systematically.

Recent work has begun to formalize evaluation of LLMs in psychiatry using standardized benchmarks. For example, efforts to map LLM outputs onto structured suicide assessment frameworks such as the Columbia–Suicide Severity Rating Scale (C-SSRS [44]) have examined whether models can classify ideation intensity, intent, and planning dimensions

in ways consistent with clinical coding. Psychiatry-specific benchmarks such as PsychBench [45] and QUEST [46] have further evaluated LLM reasoning across psychiatric diagnostic and risk-related scenarios using structured scoring rubrics and human expert adjudication. More recently, dedicated mental health LLM benchmarks including MindBench [47] and VeraMH [48] have been introduced to assess safety, clinical reasoning, and response appropriateness across sensitive domains. These developments signal a shift from informal vignette testing toward standardized, reproducible evaluation frameworks. The present study complements these approaches by focusing on multi-domain psychosocial risk and by integrating structured JSON-based assessment within a live conversational architecture rather than evaluating isolated prompt–response pairs.

Equally important are questions of trust, transparency, and safety. People's willingness to engage with AI in healthcare depends not only on technical accuracy but also on whether they perceive the system as reliable, comprehensible, and aligned with human values. Awareness that an AI system is involved can either strengthen trust by suggesting objectivity and consistency or weaken it if the system feels opaque or impersonal [49]. These concerns are reinforced by evidence of bias, hallucination, and inappropriate outputs in LLMs [50,51]. Together, these issues highlight that user trust and model safety are interdependent. Transparency, supervision, and empathetic communication are essential to both. At the same time, little research has explored how structured, protocol-driven assessments can be integrated into therapeutic conversations without undermining relational quality. Although LLMs are often praised for conversational fluency and perceived empathy [17–19], it remains unclear whether these strengths can coexist with the rigor required for clinical assessment. Addressing these questions requires systematic, multi-domain evaluations that balance accuracy with ecological validity, transparency, and relational sensitivity.

## The current study

The present research addresses these limitations through three interlinked studies that combine technical benchmarking, user validation, and functional chatbot implementation. Across all three studies, the overarching goal was to establish how well LLMs conduct clinically meaningful risk assessments before considering deployment with real users in high-stakes contexts. Study 1 evaluated the capacity of two leading LLMs—GPT-4 (OpenAI) and Claude 3 Sonnet (Anthropic)—to identify and classify risk in a standardized set of vignettes derived from participants' lived experiences across three domains: suicidality and self-harm, intimate partner violence, and substance misuse. By grounding the vignettes in anonymized survey data rather than artificial scenarios, the study provided an ecologically valid benchmark of current model performance. Furthermore, participants' own ratings of the highest level of risk they believed they had experienced were used as the primary benchmark for severity classification. Although clinician-administered tools are widely used in practice, decades of research demonstrate that even structured professional judgment has only modest predictive accuracy for suicide and other forms of psychosocial risk [52,53]. Moreover, the aim of the present study was not to predict future outcomes but to examine whether LLMs could interpret lived-experience narratives in ways that align with individuals' perceived severity at the time. Participant self-ratings therefore provided a theoretically consistent reference point for evaluating concordance between human and model interpretations, while acknowledging that such ratings are not equivalent to formal clinical diagnosis or actuarial prediction.

Study 2 extended this work by asking participants themselves to evaluate the AI-generated responses to vignettes based on their own disclosures. This allowed examination not only of technical accuracy but also of how individuals perceived the models' empathy, understanding, and usefulness—key relational factors that influence trust and engagement with digital interventions. Study 3 advanced beyond vignette-based evaluations to test a supervised chatbot system in simulated conversations. Using Amanda, a therapy-oriented LLM, as the base conversational agent, the system incorporated a second GPT-4o "AI supervisor" to monitor for risk and a separate risk assessor to conduct a structured, JSON-based protocol—a standardized format that organizes the chatbot's responses and risk-assessment items into clearly defined data fields to ensure consistency and replicability—when risk was detected. This design allowed assessment of

whether LLMs can carry out comprehensive risk assessments while maintaining a therapeutic tone and conversational flow.

## Study 1: Comparison between LLMs

Study 1 examined how well two leading LLMs, GPT-4 and Claude 3 Sonnet, could identify and assess psychosocial risk across three domains: suicidality and self-harm, IPV, and substance misuse. A total of 180 participants with relevant lived experience completed domain-specific surveys, providing structured responses about a past suicidality, IPV, or substance misuse experience. These responses were used to construct anonymized vignettes, which were then evaluated by both LLMs using standardized prompts. This study aimed to test the models' accuracy in identifying the presence and severity of psychosocial risk, using participants' self-rated risk as the benchmark. Study 1 also captured participants' own attitudes toward the use of AI in risk assessment contexts.

## Method

### Ethics statement

The study was approved by University of Lausanne ethics committee (Project ID: C_SSP_082024_00012) and applies to the subsequent studies using the same data as well and complies with the Declaration of Helsinki for human subjects research. Electronic consent was obtained from all participants by ticking a box indicating their consent in the survey.

### Participants

We recruited 180 participants to complete one of three web-based surveys focused on either suicidality and self-harm (n = 52), intimate partner violence (IPV; n = 50), or substance misuse (n = 78). The average age of participants was 38.25 years (SD = 11.24), with slightly younger participants in the suicidality/self-harm group (M = 36.56, SD = 11.10) and slightly older participants in the substance use group (M = 40.21, SD = 11.44). In the suicidality/self-harm group, most participants identified as men (71.2%), whereas the IPV group included more women (72.0%). Gender distribution in the substance misuse group was more balanced, with 42.3% identifying as men, 57.7% as women, and 3.8% selecting another gender identity. The majority of participants identified as White (65.0% overall), followed by Black (13.3%), Asian (10.0%), and mixed or other ethnic backgrounds (11.7%), with distribution varying slightly across risk groups. Educational attainment was broadly distributed: 15.6% had completed high school, 31.1% reported some college experience, 35.0% held a bachelor's degree, and 21.7% had earned a graduate or professional degree. In terms of employment status, 75.6% of participants reported working full-time, 17.8% part-time, and 6.7% were unemployed or in other categories. A full breakdown by domain is provided in Table 1.

### Procedure

Participants were recruited via the online research platform Prolific between September and October 2024 and were invited to take part in a study focused on their past experiences with one of three psychosocial risks: suicidality and self-harm, intimate partner violence (IPV), or substance misuse. Recruitment occurred through three separate surveys, each targeting a specific risk domain. Individuals were first presented with a series of prescreening items tailored to the risk category of the corresponding survey. Those who met eligibility criteria were permitted to continue with the main survey.

Each survey began with domain-specific screening questions to determine whether participants had relevant lived experience and were not currently in crisis. Participants were asked whether they had ever experienced suicidality and self-harm, IPV, or substance misuse, depending on the survey version. Those who responded affirmatively were asked a series of follow-up questions about the nature and timing of their experiences. Individuals who indicated that they were currently experiencing any of the identified risks were excluded from the study and automatically redirected to the end of the survey, where they were provided with a list of mental health and crisis support resources. Eligible participants proceeded to the main survey, which included questions assessing demographics, prior experience with AI chatbots or virtual assistants, and exposure to

**Table 1. Study 1 Demographics Across Risk Domains.**

| Variable | Suicide/Self-harm | IPV | Substance Use |
|---|---|---|---|
| | (*n* = 52) | (*n* = 50) | (*n* = 78) |
| **Age** (*M, SD*) | 36.56 (11.10) | 37.36 (10.51) | 40.21 (11.44) |
| **Gender** | | | |
| Man | 37 (71.2%) | 14 (28.0%) | 33 (42.3%) |
| Woman | 19 (36.5%) | 36 (72.0%) | 45 (57.7%) |
| Other | 0 (0.0%) | 0 (0.0%) | 3 (3.8%) |
| **Ethnicity** | | | |
| White | 30 (57.7%) | 31 (62.0%) | 56 (71.8%) |
| Black | 10 (19.2%) | 6 (12.0%) | 10 (12.8%) |
| Asian | 6 (11.5%) | 7 (14.0%) | 5 (6.4%) |
| Mixed/Other | 6 (11.5%) | 6 (12.0%) | 7 (9.0%) |
| **Education** | | | |
| High school | 9 (17.3%) | 6 (12.0%) | 13 (16.7%) |
| Some college | 17 (32.7%) | 18 (36.0%) | 21 (26.9%) |
| Bachelor's | 18 (34.6%) | 17 (34.0%) | 28 (35.9%) |
| Graduate+ | 11 (21.2%) | 9 (18.0%) | 19 (24.4%) |
| **Employment** | | | |
| Full-time | 37 (71.2%) | 40 (80.0%) | 60 (76.9%) |
| Part-time | 12 (23.1%) | 7 (14.0%) | 12 (15.4%) |
| Unemployed/Other | 4 (7.7%) | 3 (6.0%) | 6 (7.7%) |

adverse childhood experiences (ACEs). Participants then completed a structured set of multiple-choice questions about a specific past experience related to suicidality and self-harm, IPV, or substance misuse, depending on their survey condition. In the final section, participants responded to items assessing their perceptions of AI-based risk assessments, including their comfort with and trust in LLMs performing initial evaluations and offering recommendations. This procedure allowed for the collection of both lived-experience data and perspectives on the potential use of AI in assessing psychosocial risk.

## Materials

The main survey consisted of demographic information, previous chatbot experience, and detailed questions about participants' past experience with a specific psychosocial risk. All materials were presented via Qualtrics. All data and materials used in the study can be found on the Open Science Framework Project page: https://osf.io/bhdz5.

The demographics section included items assessing participants' gender identity, age, sexual orientation, ethnicity, education level, disability status, and employment status. Items allowed for inclusive response options and open-text fields where appropriate. In the chatbot experience section, participants were asked whether they had previously used AI-powered chatbots (e.g., ChatGPT, Bing, Bard) or virtual assistants (e.g., Siri, Alexa). Those who responded affirmatively were asked to indicate the specific tool used and their level of familiarity with it. An additional question asked participants to consider how strongly they believed these experiences had impacted their health (rated on a 3-point Likert scale: "Not much," "Some," "A lot").

In the risk-specific section, participants completed one of three structured questionnaires corresponding to the risk domain most relevant to their lived experience: self-harm or suicidality, intimate partner violence, or substance misuse. Each questionnaire included between 20 and 30 items and used a mix of multiple-choice, Likert-type, and open-ended formats. The questionnaires were administered online in plain, non-clinical language and were designed to balance sensitivity, clarity, and comparability across domains.

***Self-harm and suicidality.*** This questionnaire assessed age of onset, precipitating triggers, mental-health context, frequency and methods of self-harm, and perceived risk at the time. It also explored coping strategies and recovery factors. Example items included "At what age did you first engage in self-harm?" and "What was your primary intention when engaging in self-harm?" Open-ended questions invited participants to describe factors influencing perceived risk or what helped them to stop.

***Intimate partner violence***. Items covered demographic characteristics of the relationship, forms of abuse (physical, psychological, sexual, and financial), frequency and severity of incidents, help-seeking behavior, and outcomes. Example items included "What were the early signs of abuse in your relationship?" and "Did the violence ever result in injuries requiring medical attention?" Participants were also asked to describe perceived risk and factors that influenced their ability to leave or seek help.

***Substance misuse***. This section focused on substance type and method of administration, patterns of use, health and social consequences, overdose experiences, recovery efforts, and relapse triggers. Example items included "Which substances did you use during the period when you felt you needed help stopping?" and "How significantly did substance use affect your daily life?"

Across all domains, participants were asked to rate the highest level of risk they believed they had faced to their emotional or physical well-being using a four-point scale (no risk, low, medium, high). They then provided a brief open-ended explanation describing the factors that influenced their judgment. This measure was later used to compare human perceptions of risk severity with LLM-generated assessments.

## Vignette construction

Participant responses to the risk-specific surveys were used to create anonymized vignettes that served as standardized case materials for LLM evaluation. These vignettes aimed at capturing a variety of psychosocial risks across suicidality/self-harm, IPV, and substance misuse. Case details were drawn from participant questionnaire responses but written by the research team to ensure there was no identifying information, enhance readability, and maintain consistency of tone and style.

Despite the past nature of the participants' risk, each vignette was framed in the present tense to emphasize immediacy and provide a realistic basis for risk assessment. The vignettes varied in terms of demographic features (e.g., gender, age, and family structure), severity of risk (low to high), and protective factors (e.g., social support, treatment engagement), ensuring that the models were exposed to a wide range of presentations representative of real-world encounters.

The vignette development process began with an initial set of cases being drafted by researchers based on participant surveys and relevant clinical frameworks. These were then refined in collaboration with GPT-4, which was used to coordinate writing style and highlight contextual details while maintaining researcher oversight. No additional information beyond what the participants had provided in the survey were included in the vignettes. The final vignettes included contextual information such as relationship dynamics, patterns of substance use, or triggers for self-harm, thereby encouraging nuanced appraisal by the LLMs. Table 2 provides illustrative examples of vignettes drawn from each domain.

## LLM risk assessment

The second phase of the study evaluated the ability of large language models (LLMs) to assess psychosocial risk using the anonymized vignettes constructed from participant responses. Two LLMs were tested: GPT-4 (OpenAI) and Claude 3 Sonnet (Anthropic). Both models were accessed programmatically via their respective APIs—OpenAI's GPT-4 (version 20240613) via openai.ChatCompletion.create() and Claude 3 Sonnet (version 20240229) via the Anthropic API using the anthropic.HUMAN_PROMPT and anthropic.AI_PROMPT format.

Vignettes describing past substance misuse ($n = 78$), suicide/self-harm ($n = 52$), and domestic violence ($n = 50$) experiences were submitted to both models. Vignettes were input in plain-text format and standardized in structure and length to

**Table 2. Examples of Vignettes Created in Study 1.**

| Domain | Example Vignette |
|---|---|
| Suicide/ Self-harm | "Claire is a 30-year-old White-Latino lesbian woman who is presenting at a mental health clinic due to ongoing symptoms of anxiety and depression. She discloses that she has been experiencing suicidal thoughts, which have been particularly intense and frequent in recent weeks. Claire first began experiencing suicidal thoughts at the age of 13, during a period of significant life changes and family conflict when she was also bullied in school. These early struggles also marked the beginning of her battles with depression, anxiety, and PTSD. Although her mental health issues have fluctuated over time, they have persisted into her adult life and are now prompting her to seek help. Claire's suicidal thoughts are affecting her ability to form and maintain relationships, and they are becoming increasingly difficult for her to manage. She has made vague plans for suicide, including methods such as overdosing or jumping from a height or in front of a moving object. However, she has not made any attempts due to a fear of the consequences and fear of death. In addition to her suicidal thoughts, Claire reports engaging in self-harming behaviors approximately once a week. She began self-harming at the age of 16, primarily as a response to bullying at school. Her current behaviors include burning and hitting herself, pulling her hair, picking at skin/scabs, and pinching herself with needles. She explains that these actions are attempts to feel something when she feels emotionally numb. These self-harming episodes typically last between 5–15 minutes and have gone largely unnoticed by those around her." |
| Intimate Partner Violence (IPV) | "Louise, a 22-year-old white heterosexual woman, presents at a mental health clinic seeking help for her current relationship. She has been in a relationship for about nine months with a man who is four years older and of the same ethnicity. Early in the relationship, her partner began isolating her from friends and family, verbally insulting her, and displaying controlling behaviors—monitoring her movements and restricting her freedom. Over time, the abuse escalated to frequent physical aggression, including pushing, shoving, throwing objects, and spitting. Though Louise has sustained minor injuries such as bruises and scratches, she has not sought medical attention, fearing it might worsen the situation. In addition to the physical violence, Louise reports enduring sexual coercion, where her partner pressures and manipulates her into unwanted sexual acts, leading to significant emotional distress. Financially, she is forced to hand over her earnings and has faced repeated sabotage of her education and work opportunities, making her feel trapped and financially dependent on her abuser. Louise describes living in a constant state of fear and anxiety, though she does not believe her life is in immediate danger. The emotional abuse has left her feeling hopeless, depressed, and with a loss of self-esteem, but she expresses uncertainty about whether she has the resources or support to leave. Despite recognizing the toll on her mental health, she has not yet sought formal help outside of this clinic visit." |
| Substance Misuse | "Sarah, a 20-year-old bisexual white woman, presents at a counseling center seeking help for her ongoing substance misuse. She reports frequently using alcohol and cannabis, often mixing them in unsafe environments, which has led to increased risks, such as impaired judgment, aggression, and frequent accidents. While she has not experienced an overdose, Sarah acknowledges that her substance use has significantly affected her daily life—her performance at school and work has declined, though she continues to attend both. Sarah describes how her substance use has worsened her underlying depression and caused instability in her moods. She admits to occasionally stealing to fund her habits, which has contributed to further feelings of shame and guilt. Despite multiple attempts to quit, including going cold turkey, gradually reducing her usage, and attending counseling, she has struggled to maintain lasting change. Sarah is aware that her continued substance misuse is affecting her mental and physical health, but she remains conflicted about how to break the cycle of addiction. She is now seeking professional treatment and support to help regain control of her life." |

ensure consistency across model conditions. Prompt engineering was conducted prior to deployment, with iterative testing to determine the optimal phrasing and sequencing of instructions. Each model received identical input text and prompts for each vignette.

Prompts included a brief instruction explaining the LLM's task followed by a set of structured questions. These questions were designed to assess multiple domains of psychosocial risk evaluation. Example items included: identifying

whether any risk was present and specifying the type (e.g., substance misuse, suicidality, domestic violence); assessing severity (e.g., none, low, medium, high); identifying key protective factors; and proposing immediate actions or interventions. Additional items asked the models to generate empathetic and supportive responses and suggest appropriate follow-up questions related to immediate safety and treatment planning.

All responses were saved in structured JSON format and parsed into tabular form for researchers' analysis. Outputs from the two models were compared in terms of accuracy (e.g., whether the correct risk type was identified), depth of risk formulation, severity classification, and conversational quality (e.g., empathy, clarity, appropriateness of tone). These model responses were used in the third phase of the study, in which participants evaluated their quality and usefulness in a simulated clinical decision-making context.

## Results

### Risk detection accuracy

Both models showed perfect performance accuracy in identifying the primary risk domain of the vignettes presented: GPT-4 and Claude both had a 100% accuracy across all three domains, implying that both systems were able to reliably detect the intended category of psychosocial risk without making errors.

### Risk severity classification

Both models demonstrated high overall agreement with the participants' own ratings of their risk across domains, with only modest differences between them. In the suicidality and self-harm domain, GPT-4 achieved 80% agreement with the participants' own ratings ($\kappa = .84$), while Claude followed closely with 78% ($\kappa = .81$). Agreement was slightly higher for IPV vignettes, with GPT-4 reaching 88% ($\kappa = .90$) and Claude 86% ($\kappa = .88$). For substance misuse cases, GPT-4 showed 83% agreement ($\kappa = .85$), and Claude 79% ($\kappa = .82$). When averaged across all domains, GPT-4 demonstrated 84% agreement with human participants ($\kappa = .87$), while Claude showed 81% ($\kappa = .84$), suggesting broadly comparable performance between the two models.

## Discussion

Findings from Study 1 demonstrated that both GPT-4 and Claude could reliably detect the intended risk domain in all vignettes and showed high agreement with participants' self-ratings of risk severity. GPT-4 showed slightly higher accuracy than Claude across all three domains, though differences between the models were modest. Agreement was highest for IPV and lowest for suicidality and self-harm, which may reflect the difficulty LLMs have in interpreting cases where risk is described in subtle or less explicit ways. The 100% domain identification observed for both models should be interpreted as performance under structured and single-domain conditions. Because vignettes were constructed to reflect one primary risk type and did not include comorbid or deliberately ambiguous presentations, these findings represent a baseline test of categorical detection rather than a proxy for real-world diagnostic complexity. Accordingly, these findings should not be taken as evidence of robust real-world performance in uncontrolled clinical environments, where disclosures are fragmented, multi-domain, and contextually embedded. One limitation of this study is that although the vignettes were based on participants' structured responses, they were written by the research team to ensure anonymity and consistency. As a result, the vignettes may not have fully captured the subjective nuances of participants' lived experiences. This mismatch could have contributed to discrepancies between participant and LLM ratings of risk severity. Despite this, the study provides an important foundation for evaluating LLMs in more interactive and dynamic contexts, as explored in Studies 2 and 3.

## Study 2: Participants ratings of LLM responses

While Study 1 assessed the performance of LLMs in identifying and classifying psychosocial risk based on participant-generated vignettes, Study 2 sought to evaluate how individuals with lived experience perceived the quality and relevance of the chatbot responses. Specifically, this follow-up study asked participants to review model-generated

outputs for the vignette based on their own disclosures, allowing for a direct comparison between AI-generated assessments and the lived experiences they aimed to reflect. This approach provides an important complementary perspective to standard accuracy metrics by centering user evaluations of appropriateness, empathy, and perceived helpfulness—dimensions critical to the potential acceptability and integration of LLMs in mental health care settings. Study 2 therefore aimed to determine (1) how accurately participants felt the models captured their experience, (2) whether the language, tone, and suggestions provided by the models were perceived as relevant and empathetic, and (3) whether perceptions differed between models (GPT-4 and Claude) or across risk domains (suicide/self-harm, intimate partner violence, and substance misuse). In doing so, the study builds on the technical evaluation from Study 1 by incorporating a person-centered validation of AI-generated outputs.

## Method

### Participants

A total of 111 participants completed the follow-up study, which represented a subsample of the original participant pool described in Study 1. These individuals were contacted again via the Prolific platform and were invited to evaluate responses generated by LLMs using vignettes based on their own prior disclosures. Of these 111 participants, 47.75% identified as women, 30.63% as men, and 21.62% chose not to disclose their gender. In terms of racial and ethnic identity, the majority identified as White (52.25%), followed by Black or African American (10.81%), Asian (5.41%), and Hispanic or Latino (3.60%). A smaller number identified with multiracial or other ethnic categories, including combinations such as White and Hispanic or Latino, or Native Hawaiian or Pacific Islander, while 22.52% of participants did not report their ethnicity. Approximately 26.13% of participants reported having some college experience without obtaining a degree, while 22.52% held a bachelor's degree and 13.51% had earned a graduate or professional degree. Another 11.71% had completed high school or an equivalent credential, and 4.50% reported "other" forms of education. Education status was missing or unreported for 21.62% of the sample. In terms of employment, the majority of respondents (58.56%) indicated being employed in a full-time, part-time, or self-employed capacity. An additional 13.51% were unemployed, 2.70% were students, and 0.90% were retired. Employment status was not reported by 21.62% of participants.

### Procedure

In the second study, participants from Study 1 were recontacted to evaluate responses generated by a LLM to the vignette based on their own experience. Each participant was randomly assigned to view either the GPT-4 (OpenAI) or Claude (Anthropic) responses. Using each participants' Prolific ID, each LLM-generated outputs in response to the participant's vignette were extracted from a structured JSON file and embedded directly into a personalized Qualtrics survey.

Participants were shown the vignette created from their responses followed by the LLM's responses across four domains: (1) immediate actions or interventions, (2) empathy and support, (3) follow-up questions for assessing immediate risk, and (4) follow-up questions for determining appropriate treatment options. For each domain, participants were asked to respond to 3–5 quantitative items rated on 5-point Likert scales (1 = Not at all, 5 = Very much), along with optional open-ended questions inviting elaboration (for example, Why or why not?). Below, we describe the questions for each of the domains.

*Immediate actions or interventions*. Participants reviewed the LLM's recommendations for how a clinician might respond immediately to the risk scenario and rated the quality, relevance, and appropriateness of those actions. Example items included "How good do you feel the response to the question is?" and "How relevant do you feel the recommendations are?" Participants also indicated whether they agreed with the suggested actions (yes/no) and whether these were consistent with any professional advice they had previously received (yes/no/have not received advice).

*Empathy and support*. Participants rated how empathetic and supportive the LLM's message felt and whether it would have made them feel understood. Items such as "How empathetic and supportive do you feel the chatbot's response

is?" were rated on a 5-point scale, followed by a yes/no question ("Do you think the response would have made you feel understood and supported?") and an open-ended prompt for additional explanation.

**Follow-up questions for assessing immediate risk**. Participants evaluated the suitability of the model's follow-up questions for determining whether a person might be at imminent risk. Ratings focused on perceived adequacy ("How well do you feel the follow-up questions assess immediate risk?") and relevance ("How relevant do you feel the follow-up questions are for determining immediate risk?"), each using 5-point scales, with an open-ended item for qualitative comments.

**Follow-up questions for identifying treatment options.** In this final section, participants assessed how effectively the model's questions explored treatment needs and next steps. Items such as "How well do you feel the follow-up questions assess the best treatment options?" and "How relevant do you feel the follow-up questions are for determining the best treatment options?" were rated on 5-point scales, with space for additional feedback.

## Data analysis

Participant evaluations of the LLM-generated responses were analyzed using a combination of descriptive and inferential statistics. Responses were grouped by the language model condition (GPT-4 or Claude) and stratified by vignette type: suicide/self-harm ($n = 36$), IPV ($n = 39$), or substance misuse ($n = 35$). For each group, we calculated means and standard deviations across key outcome variables, including perceived accuracy of the vignette, quality and relevance of the response, empathy, and the usefulness and relevance of suggested risk questions and treatment options. These items were rated on 5-point Likert-type scales.

To compare evaluations between the two models, a series of independent samples t-tests were conducted for each outcome. In addition, proportions were calculated for binary variables assessing whether participants agreed with the model's suggested actions, felt the recommendations aligned with their past behavior, and reported feeling understood. All analyses were conducted using Python and R, and results are reported in accordance with APA 7 guidelines.

## Results

Across all participants, model-generated responses were rated moderately to highly agree on most dimensions (see Table 3 for the full results). Participants generally felt that the LLMs provided relevant and appropriate responses, with particularly favorable ratings for the relevance and quality of risk questions and suggested treatment options. Empathy and overall response quality were also rated positively, although with slightly more variability across participants. The majority of the participants agreed with the actions suggested by the LLM (86.5%) and the same percentage of participants stated that these actions were consistent with past actions taken or recommended. Around 70% of participants felt understood by the LLM. There were no significant differences in ratings between participants who viewed GPT-4 responses and those who viewed Claude responses within each vignette category (p-values were between .286 and .766). Similarly, the averaged AI evaluations did not differ substantially across vignette type (suicide/self-harm, IPV, or substance misuse). Together, these findings suggest that both models were perceived as similarly effective in formulating supportive, relevant, and clinically appropriate responses, regardless of the type of risk described.

## Discussion

Findings from Study 2 suggest that both GPT-4 and Claude produced responses that were generally well-received by participants with lived experience of psychosocial risk. The majority of participants felt that the model accurately identified the nature of the issue described, provided helpful and relevant suggestions, and conveyed empathy in its responses. Ratings did not differ significantly between the two LLMs or across different risk domains, indicating consistent perceptions of quality and utility across models and contexts. Notably, approximately 70% of participants reported feeling understood by the model, and over 86% agreed with the actions suggested, figures that point to promising levels of human acceptability of AI responses. At the same time, the finding that nearly 30% of participants did not feel understood warrants careful

**Table 3. Study 2 Results.**

| | Suicide (n = 36) | IPV (n = 39) | Substance misuse (n = 35) | Total (n = 111) |
|---|---|---|---|---|
| *Accuracy of the vignette* [a] | *3.14 (1.22)* | *3.44 (1.12)* | *2.94 (1.11)* | ***3.16 (1.17)*** |
| Quality of response | 3.75 (1.02) | 3.69 (0.86) | 3.69 (1.25) | **3.70 (1.04)** |
| Relevance of response | 3.94 (0.98) | 3.87 (0.83) | 4.00 (1.11) | **3.94 (0.97)** |
| Empathy of response | 3.83 (1.08) | 4.08 (0.87) | 3.70 (1.24) | **3.85 (1.10)** |
| Quality of risk questions | 4.06 (0.83) | 4.18 (0.79) | 3.76 (1.09) | **4.02 (0.91)** |
| Relevance of risk questions | 4.28 (0.78) | 4.26 (0.75) | 3.94 (1.06) | **4.17 (0.87)** |
| Quality of treatment options | 4.03 (0.94) | 4.15 (0.71) | 3.75 (1.14) | **3.98 (0.94)** |
| Relevance of treatment options | 4.19 (0.95 | 4.23 (0.74) | 3.91 (1.15) | **4.11 (0.95)** |
| % Agree with actions suggested | 86.1% | 89.7% | 82.9% | **86.5%** |
| % Consistent with past actions * | 92.0% | 88.5% | 78.3% | **86.5%** |
| % Felt understood | 63.9% | 74.4% | 75.8% | **70.6%** |

*Note.* For ease of reading, we have only included the total results in this table. For readers interested in the results including GPT and Claude separately, are referred to S1 Table in Supplemental Material.

* Only relevant to participants who had sought prior help.

a. Accuracy of the vignette was unrelated to the LLM being tested as the vignettes were not created by the LLMs.

consideration. In the context of psychosocial risk, this represents a clinically meaningful minority. Qualitative feedback describing responses as generic or "robotic" suggests that the presence of empathic language does not automatically translate into perceived relational attunement. While LLMs may accurately detect risk and generate supportive phrasing, perceived understanding appears to depend not only on linguistic markers of empathy but also on users' sense of authenticity and contextual depth in the interaction. In high-risk settings, such perceptions may influence trust, disclosure, and engagement with recommended actions. These findings therefore indicate that technical accuracy and empathic wording are necessary but not sufficient for clinical acceptability. While Study 1 focused on technical performance metrics, the current findings provide an important user-centered validation of LLM-generated risk assessments and reinforce the idea that such models can produce outputs that resonate with those directly affected by the issues being modelled.

## Study 3: Simulated patient-chatbot interactions

This study built on the findings from Studies 1 and 2 and tested a three-agent chatbot architecture in which Amanda (a LLM designed specifically for a clinical context) served as the base conversational agent, supported by an AI supervisor, and a dedicated risk assessment chatbot. This approach enabled us to evaluate a functional chatbot system designed to combine empathic conversation with structured risk assessment. For ethical reasons, the chatbot was not deployed directly with patients in crisis. Instead, it was tested using simulated scenarios which were based on the vignettes and participants' responses from Study 1. This approach allowed us to maintain participant safety while preserving ecological validity, offering a realistic test of how an implemented chatbot might respond to sensitive risk presentations. Study 3 therefore provided the first evaluation of a live chatbot integrating real-time supervision and structured risk assessment, tested against scenarios grounded in real participants' experiences.

## Method

### Chatbot architecture and design

The chatbot system used in Study 3 builds upon Amanda, a GPT-4o-powered voice- and text-based chatbot developed to support users in navigating relationship and mental health challenges [24,25]. Amanda engages users through

naturalistic, empathic conversation and has previously demonstrated acceptability and effectiveness across a range of psychosocial issues. For this study, the core Amanda architecture was maintained as the primary conversational agent. However, a multi-agent system was introduced to enable real-time detection and assessment of psychosocial risk. This system included three interacting agents: Amanda as the front-facing chatbot, an "AI supervisor," and a dedicated risk assessment chatbot.

Both the AI supervisor and the risk assessor were implemented using GPT-4o. The AI supervisor continuously monitored the ongoing conversation between Amanda and the user, using a set of prompt-engineered examples of risk-relevant language and scenarios to flag potential instances of suicidality, IPV, or substance misuse. Once the AI supervisor detected a possible risk, the conversation flow was redirected to the risk assessment chatbot—a separate GPT-4o agent initialized with a structured JSON-based prompt containing standardized risk assessment questions. This handoff enabled the system to shift from open-ended supportive dialogue to a more structured and clinically informed assessment process, without compromising user experience or conversational coherence. The system also includes an option to alert the researcher in cases of risk. However, this functionality was not used for this study given the cases were fictive. See Fig 1 for a graphical illustration of the multi-agent system architecture.

## Procedure

***Development of risk assessment questions for the AI risk assessment.*** We began by reviewing structured assessment questions commonly used to evaluate the severity and characteristics of suicidality, IPV, and substance misuse (see materials below for more detail). Using this foundation, we synthesized a set of risk assessment items that integrated critical dimensions across all three areas. Each risk assessment included between six and seven primary questions related to the risk with several additional questions asking for further information if required. The finalized question sets were formatted into structured JSON files and used to prompt the chatbot the chatbot to follow a structured protocol for risk assessment.

***Suicidality/self-harm***. For the risk assessment questions related to suicidality and self-harm, we identified 15 existing assessment resources, including standardized clinical tools (e.g., the Columbia-Suicide Severity Rating Scale [C-SSRS] [44]) and widely used digital or mobile-based self-assessment platforms (e.g., Wysa). Based on these established references, we created two sets of 11 questions to assess suicidality and self-harming behaviors, respectively. These questions cover commonly assessed dimensions such as the presence of harmful thoughts, their frequency, intensity, duration, triggering events, history of past attempts, and protective factors. Example questions include: "Have you recently experienced any thoughts of self-harm?", "How often do these thoughts occur? For example, do they occur very rarely, occasionally like once per month, often like several times a week, or very frequently, like almost daily?".

***Intimate partner violence***. For the risk assessment questions related to IPV, we reviewed eight tools, encompassing danger assessment instruments (e.g., the Hurt, Insult, Threaten, Scream [HITS] scale [54], screening protocols used in primary care and emergency settings (e.g., the Domestic Abuse, Stalking, Harassment, and Honor-Based Violence [DASH] checklist [55]), and app-based safety planning tools (e.g., Samaritans Self-Help). Based on commonly used assessment criteria and classical dimensions, we developed 11 questions to assess IPV. Example questions include: "Has the perpetrator recently made any specific threats against you, your children, or other family members?", "Do you currently feel safe in your relationship? Could you tell me what specific situations or actions make you feel unsafe?".

***Substance misuse***. For the risk assessment questions addressing substance use, we reviewed 11 resources, including established screening and diagnostic tools (e.g., the Alcohol Use Disorders Identification Test [AUDIT] [56]), and the Drug Abuse Screening Test [DAST] [57]),as well as app-based brief intervention modules (e.g., the Drinkaware app). These tools address patterns of use, dependency severity, and psychosocial consequences. In line

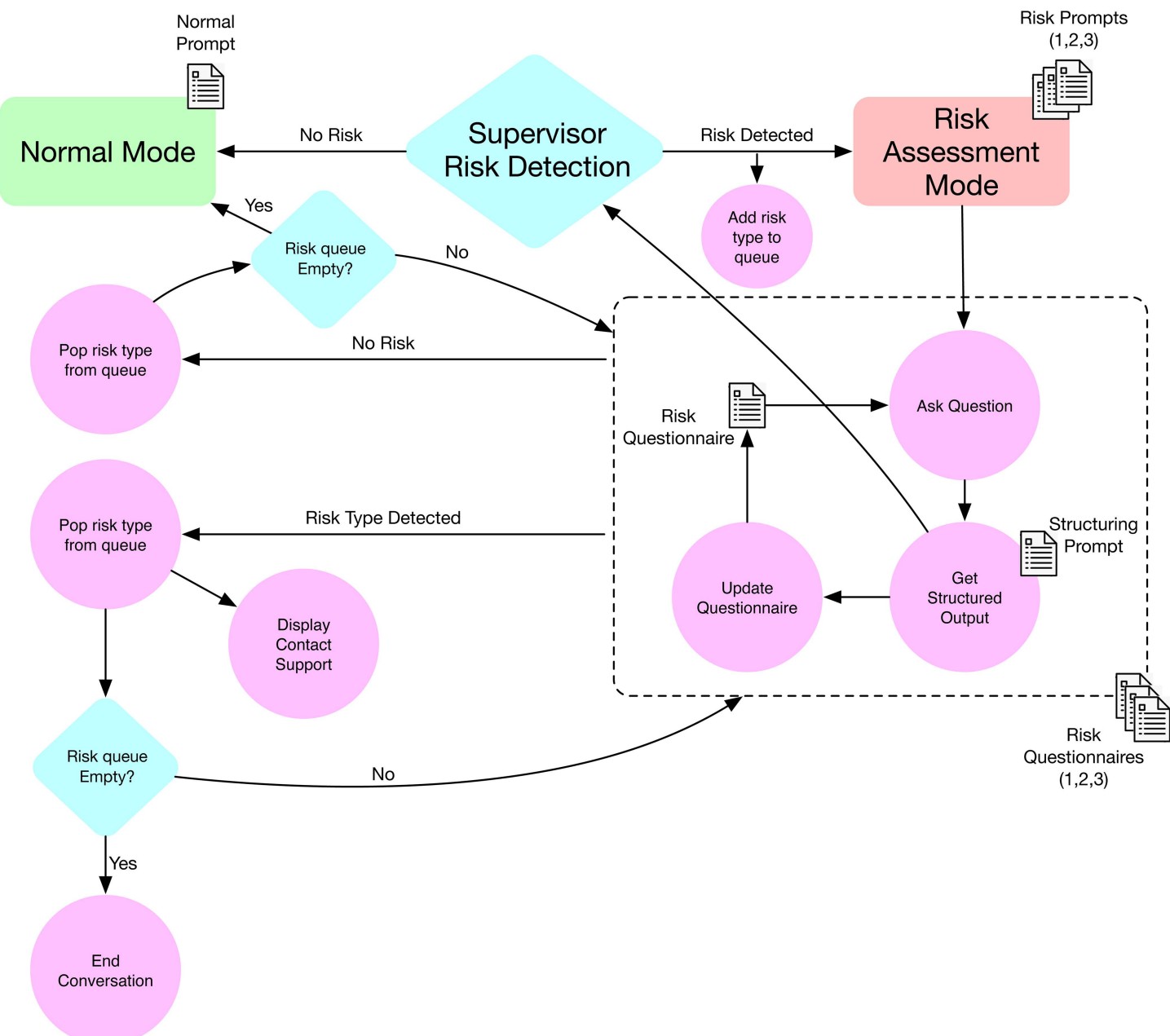

**Fig 1. The Multi-Agent System Architecture.** Note. This figure highlights the software process for handling and evaluating risk. Normal mode (top left) is the regular mode of operation for Amanda. Throughout the conversation, a separate LLM assesses for signs of risk (Supervisor Risk Detection). If any signs of risk are detected (of the types relating to suicidal ideation/self-harm, intimate partner violence, and substance misuse), we add the detected risk type(s) to a queue, and enter risk assessment mode. This mode implicates two further LLMs, one for administering the risk assessment questions, and a second for structuring the responses from the user into a questionnaire for easier downstream use. Once the questionnaire for a particular risk type is complete, the (now evaluated) risk type is popped from the queue, and the results are assessed. If it is clear that risk is present, we display relevant contact details and draw an end to the conversation. If no risk for that type is detected, we either proceed to evaluate the next risk type in the queue, or return to normal conversation mode.

with the aforementioned assessment criteria and classical dimensions, we developed 11 questions to assess substance misuse. Example questions include: "Are there any recent changes in your life that you feel are affecting your substance use, such as job loss, divorce, bereavement, or stress?", "Have you ever sought treatment or support for substance use?".

*Simulated case presentations*. We selected 40 individual cases from the Study 1 participant pool to design case presentations and example responses to the risk assessment questions designed for the chatbot. These cases included 12 self-harm or suicidality, 10 IPV, and 18 substance misuse cases. These were intentionally selected to reflect a wide variety of risks in each domain, with some case presentations representing lower-risk situations and others depicting higher levels of risk. All case presentations included only a single type of risk rather than portraying multiple risk types in one. Each case presentation included a brief description of the participant based on their reported background (e.g., demographic details, presenting problems, current risk, psychiatric history). This information was not provided to the chatbot but was given to the research assistants engaging with the chatbot to help them stay in character in case the chatbot asked questions they did not have answers prepared for.

The main part fed to the chatbot were the example responses. The example responses began by opening statements using neutral, everyday language—often describing emotional distress, interpersonal conflict, or general life stressors. This strategy allowed for a more natural progression of disclosure, with risk-related content typically emerging between the third and fifth conversational turns. This approach served two purposes: (1) to increase validity by presenting the real-life conversations, and (2) to evaluate the chatbot's sensitivity and responsiveness—specifically, whether and when it would initiate the risk assessment sequence in response to users. The example responses included statements such as "*Not direct threats, but his control over me feels like a constant threat*" in response to the question *"has the perpetrator recently made any specific threats against you, your children, or other family members?"*

*Chatbot interactions*. Using these case presentations, research assistants simulated user interactions by engaging with the chatbot one case at a time. Each assistant logged into a separate chatbot account, adopted a first-person perspective, and completed a single case before resetting the session by closing and reopening the chatbot. This procedure ensured conversation independence and eliminated contextual carryover across cases. All interaction logs were then exported for subsequent coding and analysis.

## Data analysis

To evaluate chatbot interactions in a consistent and rigorous manner, we developed a codebook to standardize the criteria for assessing the quality and completeness of human-chatbot interactions, and to increase inter-rater reliability. The codebook included two primary dimensions: interaction completeness and interaction quality. For completeness, binary coding (0 = no, 1 = yes) was used to assess: (1) Whether the conversation was completed (e.g., user or chatbot has said thank you goodbye, solution is provided); (2) Whether all risk assessment questions were asked, and chatbot provided a response to each of the risk-assessment questions. Each question was individually assessed, with coders determining whether the chatbot gave a response (coded as 0 or 1).

For interaction quality, we used eight evaluation factors which have been used to evaluate the quality of chatbot therapeutic skills [25]: (1) Error Management – handles typos, incomplete sentences, or broken language; (2) Response Length – keeps responses concise and focused; (3) Repetitiveness – avoids overly repetitive language; (4) Understanding – appropriately interprets user input and responds to the relevant content; (5) Human-Likeness – uses natural, human-like tone rather than robotic phrasing; (6) Empathy – conveys understanding and validates the user's experience; (7) Context Awareness – tailors responses using contextual knowledge; (8) Comprehensibility – uses plain, accessible language without complex terms. Each factor was assessed firstly in a binary judgment (0 = no, 1 = yes) on whether the factor was present at all. Then, coders rated the degree to which the factor was

present using a 4-point Likert scale (0 = absent, 3 = completely present). All chatbot interactions were independently evaluated by two coders.

Before coding began, both coders received training in which a trainer explained the codebook and defined each code, explaining the coding procedure, and demonstrated one example case from each of the three domains to illustrate how to apply the codebook consistently. Using these examples, coders aligned their understanding of the criteria and discussed any discrepancies. To ensure reliability, 14 cases were double coded by two independent raters first. The double-coded cases were selected to represent the full range of scenarios (five suicide or self-harm, five IPV, and four substance misuse cases) to capture variability in both interaction completeness and quality. Rater agreement levels were then calculated for both dimensions. Then, all remaining cases were divided between the two coders and coded independently, applying the finalized codebook.

## Results

**Risk assessment coverage.** The full results can be found in Table 4. Across all three domains—suicide/self-harm, IPV, and substance misuse—the chatbot successfully completed 100% of the interactions, as defined by our criteria for conversational completion: the dialogue ended with a clear closing phrase (e.g., "thank you", "goodbye"), and a solution or summary response was provided.

To further assess coverage, we examined whether each individual risk assessment item was explicitly covered during the chatbot interaction. The average item completion rate for the chatbot across domains was 100% for suicide/self-harm, 95% for IPV, and 87% for substance misuse. As some of the items were optional, these were not included in the calculation of the average completion rate. As can be seen, the chatbot asked all the required questions in the cases of suicide and self-harm. However, it missed some questions in one of the IPV cases. In this case, the participant said they did not feel safe in the relationship which prompted the chatbot to move toward trying to ensure the participant's safety rather than continuing with the risk assessment:

**Table 4. Chatbot System Completion Rates of Each Item Coded by Researchers.**

| Code | Suicidality/self-harm | | Intimate partner violence | | Substance misuse | | Rater % agreement |
|---|---|---|---|---|---|---|---|
| | % yes | % no | % yes | % no | % yes | % no | |
| Item1 | 54.55* | 45.45 | 100.00* | 0.00 | 88.89* | 11.11 | 71.4 |
| Item2 | 81.82* | 18.18 | 100.00 | 0.00 | 77.78* | 22.22 | 92.9 |
| Item3 | 100.00 | 0.00 | 20.00* | 80.00 | **83.33** | 16.67 | 92.9 |
| Item4 | 100.00 | 0.00 | 100.00 | 0.00 | **72.22** | 27.78 | 92.9 |
| Item5 | 100.00 | 0.00 | 60.00* | 40.00 | **88.89** | 11.11 | 85.7 |
| Item6 | 100.00 | 0.00 | 100.00 | 0.00 | 66.67* | 33.33 | 92.9 |
| Item7 | 63.64* | 36.36 | **90.00** | 10.00 | 100.00 | 0.00 | 92.9 |
| Item8 | 100.00 | 0.00 | **90.00** | 10.00 | **77.78** | 22.22 | 100.0 |
| Item9 | 45.45* | 54.55 | **90.00** | 10.00 | **83.33** | 16.67 | 85.7 |
| Item10 | 100.00 | 0.00 | 80.00* | 20.00 | 50.00* | 50.00 | 85.7 |
| Item11 | 90.91* | 9.09 | **90.00** | 10.00 | 100.00 | 0.00 | 100.0 |
| Item12 | NA | NA | NA | NA | 100.00* | 0.00 | 100.0 |
| **Total** | **100.00** | **0.00** | **95.00** | **5.00** | **86.51** | **13.49** | **91.1** |

*Note.* Items denoted with asterisk and italicized are items which were only asked if further information was required (e.g., Is there anything else you would like to tell me about related to this". The first question may not always have been required because the participant had already provided the answer. The items that were not required are not calculated in the total. Items that should have been asked but were not 100% of the time are bolded.

**Participant: I don't feel safe in my relationship as I am scared to do anything she may not like.**

Chatbot: I'm really sorry to hear that you're feeling unsafe in your relationship. It's very important to prioritize your safety and well-being. You've shared a lot about the difficulties you're facing, and it's understandable that you're feeling scared.

I want to make sure we're focusing on what's most helpful for you. You mentioned conflict and the feeling of not being safe. Are there any actions or resources that you think might help you feel safer, or any support you need right now? Safety is crucial, and finding ways to enhance it could be a first step toward improving your situation.

It also missed some questions across several substance misuse case presentations. In these instances, the person had said they no longer used the substances which prompted the chatbot to not ask questions about how often they used and what led to their use. For example:

**I used to consume cannabis and alcohol. However, I haven't used in the last month.**

It's really positive to hear that you've not used substances in the last month. That shows a lot of strength and determination. Since you're focusing on finding effective ways to cope, maybe you can share if there have been any recent changes or challenges in your life that you feel might be impacting your journey? This context can help in identifying the coping strategies that might work best for you.

**Not really, my life is pretty stable at the moment. However, I guess my alcohol consumption has led to some conflicts in my personal relationships.**

It sounds like you've noticed some personal relationship conflicts related to your past alcohol consumption. That's an important insight. Could you tell me a bit more about how these conflicts have affected you or your relationships? Understanding more about this can help us address any ongoing concerns and find the best strategies to support you moving forward.

In this particular example, the person had said they had not used in the past month which made the following question irrelevant. However, there were other instances where the person simply stated they no longer used but did not specify for how long. In this instance, the chatbot should have asked the follow-up questions.

Overall, the results indicate that the chatbot was highly reliable in delivering the full sequence of risk assessment content. However, it also indicates that there are occasions where the chatbot might omit certain questions thinking it had the answer already but perhaps did not or it focused on participant's immediate safety rather than completing the risk assessment.

*Interaction quality*. We next examined the quality of chatbot interactions across eight dimensions reflecting conversational competence and therapeutic appropriateness: error management, response length, repetitiveness, understanding, realism, empathy, context awareness, and comprehensibility (see Table 5). Overall, the chatbot performed consistently well, handling user input appropriately and producing coherent, contextually relevant responses. Empathy and understanding were particularly strong, with most outputs demonstrating accurate reflection of users' emotional tone and circumstances. Responses were generally clear, natural, and well-phrased, although occasional repetition and slightly mechanical wording were noted in some exchanges. Minor lapses in contextual continuity appeared primarily in the intimate partner violence and substance misuse domains but did not substantially affect overall comprehensibility. Taken together, these results indicate that the system generated interactions that were largely accurate, empathic, and intelligible across risk domains, with only minor areas for improvement in stylistic variation and sustained contextual engagement.

## Discussion

Study 3 showed that the chatbot was able to follow a structured prompt and complete a full risk assessment across most simulated scenarios. The system consistently went through most required questions and, in some cases, used additional prompts to elicit further detail. In one IPV instance, it moved onto attempting to provide immediate support for the person

**Table 5. Technical Outcomes of Interaction Transcripts Coded by Researchers.**

| Code | % yes | % no | Suicidality/ self-harm | | IPV | | Substance misuse | | Total | | |
|---|---|---|---|---|---|---|---|---|---|---|---|
| | | | Mean | SD | Mean | SD | Mean | SD | Mean | SD | Rater % agreement |
| # of interactions | | | 22.19 | 1.58 | 20.8 | 2.66 | 22.11 | 1.81 | 21.79 | 2.44 | NA |
| Error management | 100 | 0 | 3.00 | 0.00 | 3.00 | 0.00 | 3.00 | 0.00 | 3.00 | 0.00 | 100.0 |
| Response length | 90.91 | 9.09 | 3.00 | 0.00 | 2.60 | 0.70 | 3.00 | 0.00 | 2.90 | 0.38 | 85.7 |
| Repetitiveness | 100 | 0 | 1.91 | 0.30 | 2.00 | 0.00 | 2.00 | 0.00 | 1.97 | 0.16 | 92.9 |
| Understanding the response | 100 | 0 | 3.00 | 0.00 | 2.90 | 0.32 | 2.89 | 0.32 | 2.92 | 0.27 | 92.9 |
| Realism/human-likeness | 100 | 0 | 2.64 | 0.51 | 2.10 | 0.32 | 2.61 | 0.50 | 2.49 | 0.51 | 57.1 |
| Empathy | 100 | 0 | 2.73 | 0.47 | 2.90 | 0.32 | 2.78 | 0.43 | 2.79 | 0.41 | 64.3 |
| Context | 100 | 0 | 3.00 | 0.00 | 2.70 | 0.48 | 2.89 | 0.32 | 2.87 | 0.34 | 57.1 |
| Comprehensible | 100 | 0 | 2.91 | 0.30 | 2.60 | 0.52 | 2.94 | 0.24 | 2.85 | 0.37 | 78.6 |

*Note.* There was 100% agreement on binary variables. The % agreement in the table refers to the Likert-scale coding.

suggesting contacting a support system or accessing counselling. In contrast, in some substance misuse cases if the person said they no longer used, it would not ask for further detail on when the last time was and omitted certain questions related to frequency. However, it could have been important to establish when the last time someone used was and still complete the risk assessment. This suggests that while the majority of the cases the chatbot can follow a structured risk assessment, there may be times when it omits certain questions and also assumes that it has the answer when it might not have.

These failure cases are not merely minor implementation issues but illustrate a structural tension between conversational responsiveness and protocol fidelity. In prioritizing therapeutic flow or inferring that a question had already been answered, the system at times privileged relational coherence over assessment completeness. In safety-critical contexts, such assumptions could result in incomplete data capture if mandatory risk indicators are not explicitly confirmed. Future iterations should therefore incorporate explicit completeness checks or hard constraints to ensure that all required assessment items are addressed before transitioning to supportive intervention.

These results demonstrate initial feasibility of implementing a three-agent architecture in which Amanda serves as the base conversational agent, risk is monitored by an AI supervisor, and assessments are conducted through a dedicated JSON-based risk assessor. Importantly, the chatbot carried out these assessments in a therapeutic manner, with its conversational skills rated highly by evaluators. However, in the future it is important to further evaluate cases where the chatbot might fail to follow instructions to better understand when additional prompting may be necessary.

A key strength of this study lies in the use of scenarios derived from participants' lived experiences rather than researcher-created cases. This increased the ecological validity of the evaluation and offered a more realistic test of how the chatbot might perform in practice. At the same time, the reliance on simulated rather than live patient interactions remains a limitation, as real-time disclosures are often less predictable and more fragmented. Additionally, we did not include concomitant risk presentations and thus the chatbot was only asked to complete one type of risk assessment, not multiple. Future work should therefore examine deployment in controlled clinical settings, with careful monitoring of safety and user experience. Nevertheless, these findings suggest that combining conversational support with supervisory oversight and structured JSON-based risk assessment offers a promising direction for the safe and scalable use of chatbots in mental health risk evaluation.

## General discussion

Across three interlinked studies, this research systematically evaluated the ability of LLMs to conduct psychosocial risk assessment. The present studies primarily evaluate detection and structured assessment alignment under controlled

conditions and do not test prospective predictive validity or real-world clinical safety. In Study 1, both GPT-4 and Claude reliably identified the primary risk domain in all vignettes and showed high agreement with participant-rated severity, although GPT-4 performed slightly better overall. Agreement was strongest for IPV and lowest for suicidality and self-harm, highlighting the difficulty of interpreting ambiguous or indirect disclosures. These findings extend earlier work demonstrating that LLMs can detect explicit indicators of suicide risk [28] while also supporting concerns that more subtle or contextual signals may be missed [37,41]. These findings should also be interpreted in light of the growing movement toward standardized mental health benchmarks for LLM evaluation (e.g., PsychBench, QUEST, MindBench, VeraMH). Unlike these benchmarks, which primarily assess model performance on fixed prompt–response tasks or expert-scored scenarios, the present work examined multi-domain psychosocial risk using lived-experience–derived cases and extended evaluation into supervised conversational deployment. As such, our results speak to feasibility under structured, staged conditions rather than to benchmark superiority or comprehensive clinical validation.

Study 2 shifted the focus to participants' perspectives, asking individuals with lived experience to evaluate model-generated responses to vignettes based on their own disclosures. Ratings were generally positive, with most participants agreeing that the LLMs had identified the issue accurately, provided useful suggestions, and communicated in an empathetic manner. Around 70% of participants reported feeling understood, and over 86% agreed with the actions recommended by the models. Importantly, there were no differences between GPT-4 and Claude, nor across risk domains, suggesting broad applicability. These findings align with prior evidence that LLMs can generate responses perceived as empathic and supportive [17–19], and they add nuance by showing that such responses resonate with individuals whose lived experiences informed the vignettes. It is important to point out, however, that there were 30% of participants who did not feel understood by the chatbot. People who felt the chatbot did not understand them often stated that the responses sounded too generic or robotic while others said they knew a chatbot could not understand their situation. These reasons are in line with previous studies [58].

Finally, Study 3 tested a supervised, three-agent chatbot system that combined Amanda—the base conversational chatbot—with an AI supervisor and a dedicated JSON-based risk assessor. This system successfully detected potential risk, initiated structured assessments, and completed the most of the required questions, often supplementing them with clarifying prompts. Crucially, the chatbot was able to deliver these assessments in a therapeutic manner, with its empathic skills rated highly. However, when it failed, it often did so because it made an assumption the patient had answered the question already or it chose to address the immediate need (e.g., person not feeling safe). This represents the first demonstration of an LLM-driven chatbot integrating structured, protocol-based risk assessment into live conversation. By using scenarios grounded in participants' lived experiences rather than hypothetical cases, Study 3 also addressed critiques of ecological validity in prior work [41]. Together, the three studies provide converging evidence that LLMs, as a multi-agent system when properly prompted to focus on risk, can contribute meaningfully to psychosocial risk detection and evaluation, while also advancing the field by introducing a replicable, JSON-based framework for assessing their performance. It also highlights the need to more carefully address edge cases and assumptions that the chatbot might make instead of asking clarifications when a clinician might do.

## Implications for theory, research, and practice

The present research advances theoretical understanding of how psychosocial risk is communicated and evaluated in digital contexts. Our findings demonstrate that LLMs can detect both explicit and indirect risk, but variation emerges in how severity is rated, with suicidality and self-harm cases showing slightly less agreement with participant perspectives than IPV or substance misuse. This pattern reinforces theoretical work on risk communication that highlights the complexity of assessing severity when contextual or subjective elements are involved [5]. At the same time, participants often perceived the LLM responses as empathic and supportive [17–19], extending theories of human–AI interaction which suggest that people treat AI systems as social actors [26]. By embedding structured, JSON-based risk assessments within a

therapeutic dialogue, Study 3 also contributes to digital mental health frameworks by showing that formalized assessment protocols can coexist with relational interaction, bridging the gap between checklist-style tools and conversational support [59]. More broadly, this aligns with critiques of AI evaluation in healthcare, which argue that accuracy alone is insufficient and that assessments must also capture clinical safety, appropriateness, and utility [60]. By introducing a replicable, structured protocol for LLM-based risk assessment, this research responds to that challenge and offers a theoretical bridge between structured assessment frameworks and empathic conversation.

This work also makes methodological contributions that shape future research on AI in mental health contexts. First, it introduces a replicable JSON-based protocol for evaluating LLMs, offering a systematic approach to benchmark performance across domains and models. Second, it addresses concerns about ecological validity by grounding vignettes and chatbot scenarios in lived-experience data rather than artificially constructed cases. The multi-method design—from technical benchmarking in Study 1, to participant validation in Study 2, and AI (and human) supervised conversational simulation in Study 3—provides a framework for staged evaluation before patient-facing deployment.

For practice, the findings suggest that LLMs may assist with structured information gathering and preliminary triage under supervision, but they should not be used as standalone decision-makers for determining clinical risk level, discharge safety, or emergency intervention. Both GPT-4 and Claude were generally able to provide structured assessments that aligned closely with participant perspectives, while the supervised chatbot in Study 3 demonstrated how AI systems can integrate empathic communication with safety-oriented protocols. These capabilities could extend access to risk assessment in settings where clinicians are scarce, such as waitlist management, after-hours support, or underserved regions. However, the research also demonstrated how these protocols may fail under certain circumstances highlighting the importance of the risk assessments to be checked and followed up by licensed clinicians. In our chatbot, it is possible to turn on a function that sends an alert to the researcher's email if a risk is identified allowing the researcher to read the transcript and follow-up related to the risk assessment if necessary. It is important to maintain some level of human oversight in case the chatbot fails and to also follow-up with the participant if the risk assessment suggests that further follow-up is necessary. A central ethical concern is also the possibility of false reassurance. If a system underestimates severity or prematurely transitions to supportive dialogue without completing assessment, users may interpret this as confirmation that their risk is low. In safety-critical contexts, such miscalibration could delay help-seeking or reduce disclosure. Responsibility for oversight therefore cannot be delegated to the model itself; accountability must remain with clinicians, institutions, and developers who define deployment boundaries and monitoring mechanisms.

## Limitations and future directions

Several limitations should be acknowledged. Although vignettes were derived from participant surveys, they were ultimately constructed by researchers, which may have introduced a level of control and consistency that does not fully capture the nuance or emotional depth of real-world disclosures. This limitation may explain some of the discrepancies between model-rated and participant-rated severity. Furthermore, while participants' self-reports of past experiences provided valuable material for vignette development, such retrospective accounts may not fully align with actual behaviors or the complexities of clinical presentations. The reliance on Prolific samples, although diverse, also limits generalizability to clinical populations or groups outside predominantly Western contexts. In addition, Studies 1 and 2 involved static inputs rather than dynamic, real-time conversations, which constrains the extent to which findings can be generalized to open-ended interactions. Furthermore, the present studies did not include a clinician comparison group, validated screening tool benchmark, or traditional machine learning baseline, and therefore do not allow direct conclusions about whether LLMs outperform human assessors or alternative modeling approaches in psychosocial risk evaluation. Finally, Study 3 demonstrated proof-of-concept in a simulated setting but did not involve live patient use, and only two LLMs were evaluated, leaving open questions about how other models might perform.

Future research should build on these contributions in several specific ways. First, beyond benchmarking generic LLMs, work should explore whether training interventions—such as fine-tuning on annotated risk datasets or reinforcement learning with human feedback—can improve severity calibration, particularly in domains like suicidality where agreement was slightly lower. Second, studies should extend evaluation beyond single-session tasks to longitudinal contexts, examining how trust in AI systems develops or erodes over time and how models handle shifting levels of risk across repeated interactions. Third, research should test these systems in more naturalistic, real-world environments, starting with low-stakes or supervised deployment such as waitlist support or triage services, where clinicians remain the ultimate decision-makers. Fourth, greater attention should be given to cultural and linguistic diversity, as risk communication varies significantly across contexts and may challenge LLMs trained primarily on English-language data. Fifth, all models evaluated in the present research were accessed via commercial APIs (OpenAI and Anthropic), and we did not test locally hosted or open-weight models. While API-based systems currently offer state-of-the-art performance, locally deployable models may differ in capability and raise distinct privacy advantages. Future work should therefore examine whether comparable results hold using on-premise or open-source models to better evaluate the privacy–performance trade-off in clinical contexts. Finally, the structured JSON outputs developed here should be explored as potential tools for integration into clinical workflows, for example by feeding directly into electronic health records or structured assessment forms to support clinician decision-making. Such directions will help ensure that LLMs are not only accurate in controlled settings but also safe, effective, and equitable in real-world practice.

## Conclusion

Taken together, these three studies provide the most comprehensive evaluation to date of large language models' capacity to conduct psychosocial risk assessment. By combining technical benchmarking, participant validation, and the development of a supervised chatbot capable of structured, JSON-based assessments, this research demonstrates both the promise and the current limitations of LLMs in mental health contexts. The findings show that LLMs can detect risk across domains, generate responses that individuals perceive as accurate and empathic, and even complete structured assessments while maintaining a therapeutic tone. At the same time, important challenges remain, including the need for careful calibration of severity judgments, attention to ecological validity, and the development of robust safeguards for real-world use. Together, these results highlight a cautious but optimistic path forward in which LLMs can augment human-led care, providing scalable tools for triage, monitoring, and support, while reinforcing the central role of clinicians in managing risk.

### Declaration of generative AI and AI-assisted technologies in the manuscript preparation process

The authors used OpenAI's ChatGPT (GPT-5) to assist with drafting, editing, and refining sections of this manuscript. The tool was employed for text generation, synthesis, and formatting support. All outputs were critically reviewed, edited, and verified by the authors, who take full responsibility for the final content of the manuscript.

### Supporting information

**S1 Table. Participant evaluations of LLM-generated responses in Study 2 across risk domains (suicide/self-harm, intimate partner violence, and substance misuse) and models (GPT-4 and Claude).** Values represent means (standard deviations) for Likert-scale ratings assessing perceived accuracy of the vignette, quality, relevance, and empathy of responses, as well as the quality and relevance of risk assessment questions and treatment recommendations. Percentages indicate the proportion of participants who agreed with suggested actions, reported consistency with past actions, and felt understood by the model.
(DOCX)

## Author contributions

**Conceptualization:** Laura M. Vowels, Pegah Zeinoddin, Alex Elswick, Tiffany L. Marcantonio, Nathan D. Wood, Matthew J. Vowels.

**Data curation:** Laura M. Vowels.

**Formal analysis:** Laura M. Vowels, Pranika Vohra, Danyang Li, Pegah Zeinoddin.

**Funding acquisition:** Laura M. Vowels.

**Methodology:** Laura M. Vowels, Alex Elswick.

**Software:** Matthew J. Vowels.

**Supervision:** Laura M. Vowels, Danyang Li.

**Writing – original draft:** Laura M. Vowels, Pranika Vohra, Danyang Li.

**Writing – review & editing:** Pegah Zeinoddin, Alex Elswick, Tiffany L. Marcantonio, Nathan D. Wood, Matthew J. Vowels.

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
