## [Decision Letter · Decision Letter 0]

11 Jan 2026

PDIG-D-25-01031Large Language Models for Psychosocial Risk Assessment: A Multi-Method Evaluation Across Suicide, Intimate Partner Violence, and Substance MisusePLOS Digital Health Dear Dr. Vowels, Thank you for submitting your manuscript to PLOS Digital Health. After careful consideration, we feel that it has merit but does not fully meet PLOS Digital Health's publication criteria as it currently stands. Therefore, we invite you to submit a revised version of the manuscript that addresses the points raised during the review process. Please submit your revised manuscript by Mar 12 2026 11:59PM. If you will need more time than this to complete your revisions, please reply to this message or contact the journal office at digitalhealth@plos.org.  Please include the following items when submitting your revised manuscript:* A letter that responds to each point raised by the editor and reviewer(s). You should upload this letter as a separate file labeled 'Response to Reviewers '. This file does not need to include responses to any formatting updates and technical items listed in the 'Journal Requirements' section below.* A marked-up copy of your manuscript that highlights changes made to the original version. You should upload this as a separate file labeled ''. This file does not need to include responses to any formatting updates and technical items listed in the 'Journal Requirements' section below.* A marked-up copy of your manuscript that highlights changes made to the original version. You should upload this as a separate file labeled 'Revised Manuscript with Track Changes '.* An unmarked version of your revised paper without tracked changes. You should upload this as a separate file labeled ''.* An unmarked version of your revised paper without tracked changes. You should upload this as a separate file labeled 'Manuscript '. If you would like to make changes to your financial disclosure, competing interests statement, or data availability statement, please make these updates within the submission form at the time of resubmission. Guidelines for resubmitting your figure files are available below the reviewer comments at the end of this letter. We look forward to receiving your revised manuscript. Kind regards, Xiaoxia Wang Academic EditorPLOS Digital Health Baki KocaballiSection EditorPLOS Digital Health Leo Anthony CeliEditor-in-ChiefPLOS Digital Healthorcid.org/0000-0001-6712-6626 '. If you would like to make changes to your financial disclosure, competing interests statement, or data availability statement, please make these updates within the submission form at the time of resubmission. Guidelines for resubmitting your figure files are available below the reviewer comments at the end of this letter. We look forward to receiving your revised manuscript. Kind regards, Xiaoxia Wang Academic EditorPLOS Digital Health Baki KocaballiSection EditorPLOS Digital Health Leo Anthony CeliEditor-in-ChiefPLOS Digital Healthorcid.org/0000-0001-6712-6626 **Journal Requirements:** If the reviewer comments include a recommendation to cite specific previously published works, please review and evaluate these publications to determine whether they are relevant and should be cited. There is no requirement to cite these works unless the editor has indicated otherwise.   If the reviewer comments include a recommendation to cite specific previously published works, please review and evaluate these publications to determine whether they are relevant and should be cited. There is no requirement to cite these works unless the editor has indicated otherwise.  **Additional Editor Comments (if provided):** Thank you for submitting your manuscript to our journal. We have now received feedback from two expert reviewers, whose comments are appended below. Both reviewers acknowledge the importance of your work, particularly its multi-method design, ecological validity through lived-experience vignettes, and innovative multi-agent architecture for safe deployment. The reviewers have also identified several areas where the manuscript can be strengthened. Given the constructive nature of these comments and the overall potential of your study, the editorial decision is Major Revision. We invite you to address all points raised by the reviewers in a revised manuscript. Please submit a detailed point-by-point response explaining how each comment has been addressed. We believe these revisions will significantly enhance the clarity, rigor, and impact of your valuable contribution to the field of digital health and AI-assisted psychosocial assessment. We look forward to receiving your revised manuscript. Thank you for submitting your manuscript to our journal. We have now received feedback from two expert reviewers, whose comments are appended below. Both reviewers acknowledge the importance of your work, particularly its multi-method design, ecological validity through lived-experience vignettes, and innovative multi-agent architecture for safe deployment. The reviewers have also identified several areas where the manuscript can be strengthened. Given the constructive nature of these comments and the overall potential of your study, the editorial decision is Major Revision. We invite you to address all points raised by the reviewers in a revised manuscript. Please submit a detailed point-by-point response explaining how each comment has been addressed. We believe these revisions will significantly enhance the clarity, rigor, and impact of your valuable contribution to the field of digital health and AI-assisted psychosocial assessment. We look forward to receiving your revised manuscript.

Reviewer 1:

General Comments

This study explores the efficacy of large language models (LLMs) for psychosocial risk assessment across three high-stakes risks: suicidality/self-harm, intimate partner violence (IPV) and substance misuse. It expands the scope of multi-domain assessments which are used frequently in the existing literature but generally includes suicide only. The research proposes a supervised multi-agent composition to apply a role-specialized architecture providing collaborative technologies including a therapeutic agent, supervisor, and structured assessor and applying JSON schema to generate structured output about risk assessment. The use of vignettes that derive from lived-experience data, rather than entirely synthetic prompts, is a good step towards enhancing ecological validity. While these are impressive strengths of the manuscript, there are also several areas that are weak and require a revision.

Specific Comments

Introduction

According to Clay (2021), it is estimated that mental health apps number from 10,000 to 20,000. This data seems old. There have been large-scale launches of applications based on LLMs since 2022. Therefore, updating this with recent data may be a good idea.

Calling ELIZA an “AI” is technically not correct because it is just a Rule-based System. Please adjust this terminology for precision.

Study 1

Reproducibility: The specific version of the Claude model used must be disclosed. To ensure reproducibility, please provide full prompt texts, model versions/dates, temperature and decoding settings, and any system/role prompts used across the studies.

What standardize instruments or expert adjudication (if any) were use to validate the severity labels? If none were used, please explain why participant self-rating was chosen rather than clinician ratings or established validated scales.

Bias Control: During the development of the vignettes, each case has been drafted by researchers according to the data provided by participants. Those researchers who used these stories are biased. How was the potential inter-researcher bias dealt with by the team?

Study 2

The OpenAI API and the Anthropic API are used in the study. It does not test any local models. Local models may perform worse but are better for patient privacy. There is a need to test whether the results hold using local models to evaluate the privacy-generalization trade-off.

The presently available information is barely adequate for comparing the two models. I recommend the use of certain statistical methods, like Semantic Textual Similarity, to more firmly prove that the outputs of the models are not different.

Another evaluation metric or benchmark would ensure it is easier to interpret and compare Study 2.

Study 3

Forcing tokens to output strict JSON formats can cause failures in LLM handling. How did the researchers deal with “bad JSON” or parsing errors?

The system's completion of the assessments "Most of the time" is not sufficiently specific. The manuscript on research project lacks numbers. Such as completion rate, detection latency, assessment coverage and safety violation rate.

General Discussion

The relation work section needs to discuss more in depth previously conducted standardized, structured suicide assessments (with LLMs), psychiatry-specific benchmarks and human evaluation (e.g. C-SSRS mapping, PsychiatryBench, QUEST)

The study does not include a clinician baseline and does not compare to any validated screening tool. It fails to match against traditional Machine Learning baselines or fine-tuned encoder/decoder models. The least that the discussion should do is show or argue that LLMs do better than machine learning on these tasks.

Reviewer 2:

This manuscript addresses a highly topical and important question: whether large language models can meaningfully contribute to psychosocial risk assessment across suicidality, intimate partner violence, and substance misuse. Overall, the paper is ambitious, carefully executed, and clearly written, and it goes well beyond much of the existing work in this area, which tends to rely on simplified prompts or single-domain evaluations. The multi-method design, in particular, is a real strength, and the authors should be commended for attempting to bridge technical performance, user perceptions, and system-level implementation within a single programme of research.

One of the most valuable aspects of the paper is its grounding in lived experience. Constructing vignettes from participants’ own disclosures, rather than from hypothetical or researcher-generated cases, substantially improves ecological validity and allows for a more realistic assessment of what current LLMs can and cannot do. Likewise, the decision to include IPV and substance misuse alongside suicidality is welcome, as it reflects the complexity of real clinical practice and avoids the narrow focus that characterises much of the existing literature.

That said, there are several areas where the manuscript would benefit from a more cautious and critical treatment of its findings. In Study 1, the reported 100% accuracy for domain detection by both GPT-4 and Claude is striking, but also raises questions about how challenging the task actually was. Given the way the vignettes were constructed and prompted, it seems likely that domain identification was relatively explicit. This does not undermine the study, but it does mean that these results should be interpreted carefully and framed as a baseline rather than as evidence of robust real-world performance. Relatedly, the use of participants’ own ratings of risk severity as the comparison standard is understandable given the study aims, but it is not equivalent to a clinical benchmark. The paper would be stronger if this limitation were acknowledged more explicitly, and if the implications of relying on subjective self-assessment were discussed in greater depth.

Study 2 is, in my view, one of the strongest parts of the manuscript. Bringing participants back to evaluate the AI-generated responses adds an important human perspective that is often missing from evaluations of mental health chatbots. The generally positive ratings for empathy, relevance, and usefulness are encouraging. However, I think the authors underplay one important finding here: around 30% of participants did not feel understood by the chatbot. In contexts involving psychosocial risk, this is not a trivial minority. Several participants reportedly described the responses as generic or “robotic”, or noted that a chatbot could not truly understand their situation. These comments deserve more analytical attention, as they point to the limits of perceived empathy and the potential consequences for trust and engagement.

Study 3 is clearly the most innovative component of the paper and represents a meaningful step forward in thinking about how LLMs might be deployed safely. The multi-agent architecture, with a supervisory model and a structured JSON-based assessor, is well motivated and thoughtfully designed. The findings suggest that such a system can, in most cases, complete structured risk assessments while maintaining a reasonably therapeutic tone. At the same time, the results also highlight important failure modes, particularly when the system assumes that certain questions have already been answered or shifts prematurely to supportive action at the expense of assessment completeness. These moments are arguably where the most clinically relevant insights lie, and I would encourage the authors to engage with them more critically rather than treating them primarily as minor implementation issues.

Across the paper, the discussion sometimes comes close to overstating what the findings can support. The models show good agreement with participants and are perceived as helpful, but this should not be conflated with clinical adequacy or safety in real-world deployment. A clearer distinction between risk detection, risk formulation, and risk prediction would help here, as would a more explicit statement about what these systems should not be used for. The ethical discussion is thoughtful, but it could be strengthened by more direct engagement with issues of responsibility, false reassurance, and the potential consequences of missed or underestimated risk.

From a presentation standpoint, the manuscript is generally well organised and readable. Some sections of the introduction are longer than necessary and could be tightened, but this is a relatively minor issue. The open science practices are a clear strength, and the availability of materials and data will be valuable for the field.

In summary, this is a strong and important paper that makes a genuine contribution to research on LLMs in mental health and psychosocial risk assessment. However, the interpretation of the results would benefit from greater restraint, and the limitations—particularly around benchmarks, task difficulty, and failure cases—need to be more fully acknowledged and integrated into the discussion. For these reasons, I would recommend major revisions. With these revisions, the manuscript has the potential to become a highly influential reference in this rapidly developing area.**Reviewers' Comments:** Reviewer's Responses to Questions Reviewer's Responses to Questions

**Comments to the Author**

1. Does this manuscript meet PLOS Digital Health’s publication criteria ? Is the manuscript technically sound, and do the data support the conclusions? The manuscript must describe methodologically and ethically rigorous research with conclusions that are appropriately drawn based on the data presented.? Is the manuscript technically sound, and do the data support the conclusions? The manuscript must describe methodologically and ethically rigorous research with conclusions that are appropriately drawn based on the data presented.

Reviewer #1: Yes

Reviewer #2: No

Reviewer #3: Yes

2. Has the statistical analysis been performed appropriately and rigorously?

Reviewer #1: Yes

Reviewer #2: Yes

Reviewer #3: Yes

3. Have the authors made all data underlying the findings in their manuscript fully available (please refer to the Data Availability Statement at the start of the manuscript PDF file)?

The PLOS Data policy requires authors to make all data underlying the findings described in their manuscript fully available without restriction, with rare exception. The data should be provided as part of the manuscript or its supporting information, or deposited to a public repository. For example, in addition to summary statistics, the data points behind means, medians and variance measures should be available. If there are restrictions on publicly sharing data—e.g. participant privacy or use of data from a third party—those must be specified.requires authors to make all data underlying the findings described in their manuscript fully available without restriction, with rare exception. The data should be provided as part of the manuscript or its supporting information, or deposited to a public repository. For example, in addition to summary statistics, the data points behind means, medians and variance measures should be available. If there are restrictions on publicly sharing data—e.g. participant privacy or use of data from a third party—those must be specified.

Reviewer #1: Yes

Reviewer #2: Yes

Reviewer #3: Yes

4. Is the manuscript presented in an intelligible fashion and written in standard English?

Reviewer #1: Yes

Reviewer #2: No

Reviewer #3: Yes

5. Review Comments to the Author

Reviewer #1: Reviewer Comments to the Author

General Comments

This study explores the efficacy of large language models (LLMs) for psychosocial risk assessment across three high-stakes risks: suicidality/self-harm, intimate partner violence (IPV) and substance misuse. It expands the scope of multi-domain assessments which are used frequently in the existing literature but generally includes suicide only. The research proposes a supervised multi-agent composition to apply a role-specialized architecture providing collaborative technologies including a therapeutic agent, supervisor, and structured assessor and applying JSON schema to generate structured output about risk assessment. The use of vignettes that derive from lived-experience data, rather than entirely synthetic prompts, is a good step towards enhancing ecological validity. While these are impressive strengths of the manuscript, there are also several areas that are weak and require a revision.

Specific Comments

Introduction

According to Clay (2021), it is estimated that mental health apps number from 10,000 to 20,000. This data seems old. There have been large-scale launches of applications based on LLMs since 2022. Therefore, updating this with recent data may be a good idea.

Calling ELIZA an “AI” is technically not correct because it is just a Rule-based System. Please adjust this terminology for precision.

Study 1

Reproducibility: The specific version of the Claude model used must be disclosed. To ensure reproducibility, please provide full prompt texts, model versions/dates, temperature and decoding settings, and any system/role prompts used across the studies.

What standardize instruments or expert adjudication (if any) were use to validate the severity labels? If none were used, please explain why participant self-rating was chosen rather than clinician ratings or established validated scales.

Bias Control: During the development of the vignettes, each case has been drafted by researchers according to the data provided by participants. Those researchers who used these stories are biased. How was the potential inter-researcher bias dealt with by the team?

Study 2

The OpenAI API and the Anthropic API are used in the study. It does not test any local models. Local models may perform worse but are better for patient privacy. There is a need to test whether the results hold using local models to evaluate the privacy-generalization trade-off.

The presently available information is barely adequate for comparing the two models. I recommend the use of certain statistical methods, like Semantic Textual Similarity, to more firmly prove that the outputs of the models are not different.

Another evaluation metric or benchmark would ensure it is easier to interpret and compare Study 2.

Study 3

Forcing tokens to output strict JSON formats can cause failures in LLM handling. How did the researchers deal with “bad JSON” or parsing errors?

The system's completion of the assessments "Most of the time" is not sufficiently specific. The manuscript on research project lacks numbers. Such as completion rate, detection latency, assessment coverage and safety violation rate.

General Discussion

The relation work section needs to discuss more in depth previously conducted standardized, structured suicide assessments (with LLMs), psychiatry-specific benchmarks and human evaluation (e.g. C-SSRS mapping, PsychiatryBench, QUEST)

The study does not include a clinician baseline and does not compare to any validated screening tool. It fails to match against traditional Machine Learning baselines or fine-tuned encoder/decoder models. The least that the discussion should do is show or argue that LLMs do better than machine learning on these tasks.

Reviewer #2: This work evaluates whether large language models (LLMs) can support scalable psychosocial risk assessment across suicide, intimate partner violence, and substance misuse. Across the three studies, GPT-4, Claude, and a supervised three-agent GPT-4o system showed strong accuracy in detecting psychosocial risk and aligning with participant-rated severity despite challenges with suicidality, while participants rated LLM responses as accurate, empathic, and clinically useful, and the chatbot reliably completed structured assessments using a JSON-based protocol. Overall, the findings show that LLMs can meaningfully aid psychosocial risk evaluation while revealing both their potential and their current limitations in mental health care.

These are several comments for the manscipt below:

1. Please change all possessive case such as participants’ lived experiences, participants’ gender identity, researchers’ analysisand participants’ perceptions.They may look like AI writing shallow rewrite by the authors.

2. The authors used Amanda to build the agents. It is not a popular and powerul tool to build LLM agents. Therefore, I recommend that the authors instead implement the framework in Python, which is standard and more robust for constructing LLM agents.

This manuscript presents research focused more on psychosocial issues and related solutions than on medical health, and I recommend transferring it to PLOS Mental Health.

Reviewer #3: This manuscript addresses a highly topical and important question: whether large language models can meaningfully contribute to psychosocial risk assessment across suicidality, intimate partner violence, and substance misuse. Overall, the paper is ambitious, carefully executed, and clearly written, and it goes well beyond much of the existing work in this area, which tends to rely on simplified prompts or single-domain evaluations. The multi-method design, in particular, is a real strength, and the authors should be commended for attempting to bridge technical performance, user perceptions, and system-level implementation within a single programme of research.

One of the most valuable aspects of the paper is its grounding in lived experience. Constructing vignettes from participants’ own disclosures, rather than from hypothetical or researcher-generated cases, substantially improves ecological validity and allows for a more realistic assessment of what current LLMs can and cannot do. Likewise, the decision to include IPV and substance misuse alongside suicidality is welcome, as it reflects the complexity of real clinical practice and avoids the narrow focus that characterises much of the existing literature.

That said, there are several areas where the manuscript would benefit from a more cautious and critical treatment of its findings. In Study 1, the reported 100% accuracy for domain detection by both GPT-4 and Claude is striking, but also raises questions about how challenging the task actually was. Given the way the vignettes were constructed and prompted, it seems likely that domain identification was relatively explicit. This does not undermine the study, but it does mean that these results should be interpreted carefully and framed as a baseline rather than as evidence of robust real-world performance. Relatedly, the use of participants’ own ratings of risk severity as the comparison standard is understandable given the study aims, but it is not equivalent to a clinical benchmark. The paper would be stronger if this limitation were acknowledged more explicitly, and if the implications of relying on subjective self-assessment were discussed in greater depth.

Study 2 is, in my view, one of the strongest parts of the manuscript. Bringing participants back to evaluate the AI-generated responses adds an important human perspective that is often missing from evaluations of mental health chatbots. The generally positive ratings for empathy, relevance, and usefulness are encouraging. However, I think the authors underplay one important finding here: around 30% of participants did not feel understood by the chatbot. In contexts involving psychosocial risk, this is not a trivial minority. Several participants reportedly described the responses as generic or “robotic”, or noted that a chatbot could not truly understand their situation. These comments deserve more analytical attention, as they point to the limits of perceived empathy and the potential consequences for trust and engagement.

Study 3 is clearly the most innovative component of the paper and represents a meaningful step forward in thinking about how LLMs might be deployed safely. The multi-agent architecture, with a supervisory model and a structured JSON-based assessor, is well motivated and thoughtfully designed. The findings suggest that such a system can, in most cases, complete structured risk assessments while maintaining a reasonably therapeutic tone. At the same time, the results also highlight important failure modes, particularly when the system assumes that certain questions have already been answered or shifts prematurely to supportive action at the expense of assessment completeness. These moments are arguably where the most clinically relevant insights lie, and I would encourage the authors to engage with them more critically rather than treating them primarily as minor implementation issues.

Across the paper, the discussion sometimes comes close to overstating what the findings can support. The models show good agreement with participants and are perceived as helpful, but this should not be conflated with clinical adequacy or safety in real-world deployment. A clearer distinction between risk detection, risk formulation, and risk prediction would help here, as would a more explicit statement about what these systems should not be used for. The ethical discussion is thoughtful, but it could be strengthened by more direct engagement with issues of responsibility, false reassurance, and the potential consequences of missed or underestimated risk.

From a presentation standpoint, the manuscript is generally well organised and readable. Some sections of the introduction are longer than necessary and could be tightened, but this is a relatively minor issue. The open science practices are a clear strength, and the availability of materials and data will be valuable for the field.

In summary, this is a strong and important paper that makes a genuine contribution to research on LLMs in mental health and psychosocial risk assessment. However, the interpretation of the results would benefit from greater restraint, and the limitations—particularly around benchmarks, task difficulty, and failure cases—need to be more fully acknowledged and integrated into the discussion. For these reasons, I would recommend major revisions. With these revisions, the manuscript has the potential to become a highly influential reference in this rapidly developing area.

6. PLOS authors have the option to publish the peer review history of their article (what does this mean? ). If published, this will include your full peer review and any attached files.). If published, this will include your full peer review and any attached files.

**Do you want your identity to be public for this peer review?** For information about this choice, including consent withdrawal, please see our Privacy Policy ..

Reviewer #1: **Yes:** Yuexiong YiYuexiong Yi

Reviewer #2: No

Reviewer #3: **Yes:** João Miguel Alves FerreiraJoão Miguel Alves Ferreira

  **Figure resubmission:** While revising your submission, we strongly recommend that you use PLOS’s NAAS tool (https://ngplosjournals.pagemajik.ai/artanalysis) to test your figure files. NAAS can convert your figure files to the TIFF file type and meet basic requirements (such as print size, resolution), or provide you with a report on issues that do not meet our requirements and that NAAS cannot fix. While revising your submission, we strongly recommend that you use PLOS’s NAAS tool (https://ngplosjournals.pagemajik.ai/artanalysis) to test your figure files. NAAS can convert your figure files to the TIFF file type and meet basic requirements (such as print size, resolution), or provide you with a report on issues that do not meet our requirements and that NAAS cannot fix.

After uploading your figures to PLOS’s NAAS tool - https://ngplosjournals.pagemajik.ai/artanalysis, NAAS will process the files provided and display the results in the "Uploaded Files" section of the page as the processing is complete. If the uploaded figures meet our requirements (or NAAS is able to fix the files to meet our requirements), the figure will be marked as "fixed" above. If NAAS is unable to fix the files, a red "failed" label will appear above. When NAAS has confirmed that the figure files meet our requirements, please download the file via the download option, and include these NAAS processed figure files when submitting your revised manuscript. **Reproducibility:** To enhance the reproducibility of your results, we recommend that authors of applicable studies deposit laboratory protocols in protocols.io, where a protocol can be assigned its own identifier (DOI) such that it can be cited independently in the future. Additionally, PLOS ONE offers an option to publish peer-reviewed clinical study protocols. Read more information on sharing protocols at https://plos.org/protocols?utm_medium=editorial-email&utm_source=authorletters&utm_campaign=protocols To enhance the reproducibility of your results, we recommend that authors of applicable studies deposit laboratory protocols in protocols.io, where a protocol can be assigned its own identifier (DOI) such that it can be cited independently in the future. Additionally, PLOS ONE offers an option to publish peer-reviewed clinical study protocols. Read more information on sharing protocols at https://plos.org/protocols?utm_medium=editorial-email&utm_source=authorletters&utm_campaign=protocols

---

## [Editor Report · Decision Letter 1]

21 Mar 2026

Large Language Models for Psychosocial Risk Assessment: A Multi-Method Evaluation Across Suicide, Intimate Partner Violence, and Substance Misuse

PDIG-D-25-01031R1

Dear Dr Vowels,

We are pleased to inform you that your manuscript 'Large Language Models for Psychosocial Risk Assessment: A Multi-Method Evaluation Across Suicide, Intimate Partner Violence, and Substance Misuse' has been provisionally accepted for publication in PLOS Digital Health.

Best regards,

Xiaoxia Wang, Ph.D

Academic Editor

PLOS Digital Health